# NeuroSuitUp: System Architecture and Validation of a Motor Rehabilitation Wearable Robotics and Serious Game Platform

**DOI:** 10.3390/s23063281

**Published:** 2023-03-20

**Authors:** Konstantinos Mitsopoulos, Vasiliki Fiska, Konstantinos Tagaras, Athanasios Papias, Panagiotis Antoniou, Konstantinos Nizamis, Konstantinos Kasimis, Paschalina-Danai Sarra, Diamanto Mylopoulou, Theodore Savvidis, Apostolos Praftsiotis, Athanasios Arvanitidis, George Lyssas, Konstantinos Chasapis, Alexandros Moraitopoulos, Alexander Astaras, Panagiotis D. Bamidis, Alkinoos Athanasiou

**Affiliations:** 1Medical Physics & Digital Innovation Laboratory, School of Medicine, Faculty of Health Sciences, Aristotle University of Thessaloniki, 54124 Thessaloniki, Greece; 2Department of Design, Production and Management, University of Twente, 7522 NB Enschede, The Netherlands; 3Department of Physiotherapy, International Hellenic University, 57400 Thessaloniki, Greece; 4Department of Computer Science, American College of Thessaloniki, 55535 Thessaloniki, Greece

**Keywords:** body–machine interface, human–robot interaction, neural rehabilitation, robotic glove, robotic jacket, serious game, soft robotics, spinal cord injury, stroke, wearable robotics

## Abstract

Background: This article presents the system architecture and validation of the NeuroSuitUp body–machine interface (BMI). The platform consists of wearable robotics jacket and gloves in combination with a serious game application for self-paced neurorehabilitation in spinal cord injury and chronic stroke. Methods: The wearable robotics implement a sensor layer, to approximate kinematic chain segment orientation, and an actuation layer. Sensors consist of commercial magnetic, angular rate and gravity (MARG), surface electromyography (sEMG), and flex sensors, while actuation is achieved through electrical muscle stimulation (EMS) and pneumatic actuators. On-board electronics connect to a Robot Operating System environment-based parser/controller and to a Unity-based live avatar representation game. BMI subsystems validation was performed using exercises through a Stereoscopic camera Computer Vision approach for the jacket and through multiple grip activities for the glove. Ten healthy subjects participated in system validation trials, performing three arm and three hand exercises (each 10 motor task trials) and completing user experience questionnaires. Results: Acceptable correlation was observed in 23/30 arm exercises performed with the jacket. No significant differences in glove sensor data during actuation state were observed. No difficulty to use, discomfort, or negative robotics perception were reported. Conclusions: Subsequent design improvements will implement additional absolute orientation sensors, MARG/EMG based biofeedback to the game, improved immersion through Augmented Reality and improvements towards system robustness.

## 1. Introduction

### 1.1. Background

Recent technological advances, innovations, and newly gained insight into the neurophysiology of motor disability have allowed for the exploration of novel research methodologies and for synergies of converging technologies in the field of neural rehabilitation [1]. The World Health Organization (WHO) has defined rehabilitation as “a set of interventions designed to optimize functioning and reduce the disability in individuals with health conditions in interaction with their environment” [2]. Neural rehabilitation, in particular, aims to recruit neural plasticity, which refers to the regeneration, repair, and reorganization of neural systems, as well as to promote behavioural principles that may help disabled patients regain function. It has been estimated that 1 out of 6 people worldwide have suffered from significant disability and about 2.4 billion individuals are currently living with a health condition that could benefit from rehabilitation [2], including neurotrauma, stroke, neoplasms, degenerative disorders, infections, congenital diseases, peripheral nervous system conditions, and neuromuscular disorders. Among the major causes of motor disability, stroke and spinal cord injury (SCI) can be considered to be of special interest. About 101 million patients with chronic stroke worldwide are facing aftermaths that affect their lifestyle [3], and stroke remains the leading cause for chronic disability in US and Europe [4]. Meanwhile, SCI impacts 0.32–0.64 million people each year worldwide, affecting young individuals in particular, while causing long-term and often irreversible disability [5].

### 1.2. Rehabilitation and Promoting Neural Plasticity

The rehabilitation process requires cross-disciplinary coordination and specialized services for comprehensive management [6]. Although the mechanisms of central nervous system reorganization through neural plasticity are not fully understood, they entail a dynamic response to environmental changes, either beneficial (adaptive) or in some situations detrimental (maladaptive) [7]. This ability of the nervous system is used in the operant learning approach of biofeedback techniques, which are based on modulating physiological parameters and providing a feedback response [8]. To that end, wearable robotic devices can be considered valuable tools for clinical or ubiquitous rehabilitation, using biosensors and allowing for continuous monitoring of physiological parameters to improve movement patterns, force, and neurological function [1,9]. Biofeedback and neurofeedback can be delivered through game-based applications, with the added benefit of increasing patient motivation and long-term engagement [10]. Participation in serious gaming and in virtual environments stimulates interactive activities and movement repetition. Especially in exercise-based games, player interaction by physically moving their bodies to play promotes motor learning through biofeedback-induced neural plasticity; this seems to have a positive and long-lasting effect on motor and functional outcomes [11]. Moreover, ubiquitous serious gaming seems to increase independence in activities of daily living (ADLs) and to improve patients’ social skills [11].

### 1.3. Robotics and Synergies in Rehabilitation

The use of robotics in neurorehabilitation is not a new concept (as it was already proposed at the beginning of the previous century [12]) and many of them are nowadays used in standard rehabilitation practice [13,14,15,16,17]. Such devices combine organically with the concepts of neuroplasticity and serious gaming, resulting in numerous successful examples of application both for the upper and the lower limbs [18,19] and for diseases spanning from trauma to muscular dystrophies and even mental disorders [1]. Rehabilitation robots (prosthetics or exoskeletons) can assist physiotherapists in driving rehabilitation through neuroplasticity, boasting some distinct advantages. Those include, among others, easy interfacing with biosensors that can in real time extract data from various sources (nerves, brain, muscles, etc.) [1], reducing the workload of physiotherapists and allowing them to offer high quality care to more patients [20], and offering repetitive, high intensity, and task specific training that can be translated to functional movements [21]. It is apparent that rehabilitation robotics is a multidisciplinary field [22] that can benefit from a systems point of view and the use of systems architecture in order to show inter-dependencies with other emerging technologies such as sensors, games, artificial intelligence, virtual reality, as well as other devices [1].

Currently in the rehabilitation of people with SCI and stroke, a number of neurorobotic technologies has been tested and have been used, with notable examples, which include localized rehabilitation procedures using robotic braces [23,24], and more large scaled explorations on combined motor restoration through orthotics [25]. Combining robotics with serious games is considered to motivate patients and to optimize motor learning [26]. However, several challenges have limited the clinical endorsement of such approaches. Among the main issues identified in the literature lies the exploratory combination of multiple rehabilitation technologies in platforms and systems, without proper system architecture or without systematic consideration of the interfaces and synergistic value between those technologies [1].

### 1.4. The NeuroSuitUp Approach

The current paper aims to present the system architecture, development endeavour, and validation experiments of a platform of wearable robotic devices and serious gaming application, aimed at the motor rehabilitation of patients with neurological disability, mainly due to SCI and stroke. In the work described hereby, we propose and validate the NeuroSuitUp platform to enable complete upper body rehabilitation, for the first time to the authors’ best knowledge, implementing wearable robotics jacket, gloves, and serious gaming. We decompose, explain, and validate the complete system with respect to hardware, sensors, actuators, energy, communication, processing, and Serious Gaming application. The motivation behind our work lies primarily with taking a systematic approach towards the development of a multi-modal platform of modular devices and applications [27], where all the underlying technologies (sensors, actuators, software, battery, and visualisation), as well as their optimal cooperation is taken into consideration a priori. Furthermore, instead of ad hoc presentation of experiments, we aimed to have the platform validated and tested under the scope of an overall unifying system architecture. The aforementioned platform will primarily be utilised in two intervention-based clinical trials for SCI [28] and stroke [29], while it is also planned to be tested in other common neurological conditions in the future, such as neurodegenerative disorders (Parkinson’s disease) and dystonia, as well as in rehabilitation of myoskeletal injuries.

The remainder of this paper is organised as follows: in Section 2 “Materials and Methods”, we first present the overlying system architecture of the NeuroSuitUp platform. We then elaborately describe the Hardware layer, including the on-board power supply unit, the sensors (Magnetic, Angular Rate, and Gravity (MARG), surface Electromyography (sEMG), flex and pressure sensors), the actuators (Electrical Muscle Stimulation (EMS) and pneumatics), the Middleware layer and the Application layer (the serious game). In Section 3 “System Validation”, we first describe the procedure and metrics used during the validation of the on-board MARG sensors through the use of a stereoscopic vision camera. We then similarly describe the validation of the soft-robotic gloves through activities of daily living (ADL) tasks. Following, we describe the participants, the questionnaires used, and the statistical analysis of our data. In Section 4, “Results”, we succinctly present the results of the validation experiments for both the jacket and the glove, as well as responses to the user experience questionnaires. In Section 5, “Discussion”, we first present and comment upon the key findings of our validation experiments, focusing on the limitations identified and on the next development plans according to the lessons learned. We then discuss our system approach by comparison to current and previous technologies and to other devices in the literature, as well as we also expand upon the clinical scope of the technologies used in our approach. Finally, in Section 6, “Conclusions”, we summarize the rationale, the validation, the results, and the conclusions of our work.

## 2. Materials and Methods

The NeuroSuitUp platform was designed to facilitate the monitoring of the kinematic chain of the upper body and the muscle activity that actuates it. For the purpose of neurorehabilitation, the modality acts as an interface between patients with SCI and a Serious Game application (Figure 1), developed in the Unity game engine, and it provides assistance in the form of Electrical Muscle Stimulation, in the relevant muscle groups. The main system is composed of a textile base, where the sensors, actuators, and processing units are located. Furthermore, a soft robotic glove (SRG) prototype was developed, as both an extension of the platform in the field of hand manipulation, and as part of the exploration of alternative actuation methodologies, specifically pneumatic, soft-robotic actuators.

The proposed SRG was developed by aiming at the human hand neural rehabilitation. The current prototype consists of a multi-sensor subsystem (MSS) and a pneumatic actuation subsystem (PAS). The NeuroSuitUp system’s middleware facilitates communication between these subsystems.

### 2.1. Hardware Layer

The wearable robotic jacket is composed of 6 Adafruit Magnetic, Angular rate, and Gravity (MARG) sensors, located at the major segments of the upper-body kinematic chain, and 10 Adafruit surface ElectroMyoGraphy (sEMG) sensors that monitor the main muscle groups of said kinematic chain. In order to ensure reliability and data transfer speed, and reduce signal loss and electromagnetic interference [30], the on-board sensors are connected in a wired configuration. The data is communicated to the rest of the system through the use of two Arduino Mega 2560s, (handling the analogue outputs of the sEMG sensors), and an Arduino Nano, (for the digital signal output of the MARG sensors) extended by an I^2^C multiplexer (TCA9548A 1-to-8 breakout board), and are then forwarded to the ROS subsystem, via Universal Serial bus (USB), running on an Ubuntu computer.

The hardware components comprising the prototype SRG are sewn, attached, and connected on an F.F. Group Work Antistatic Glove. When the PAS is activated, compressed air flows from an Air Pump and Vacuum 1.8 LPM—4.5 V DC Motor through a 6 V 2-Position 3-Way Solenoid Air Valve, controlled by an Arduino UNO R3 microcontroller, to the actuators through silicone tubes. This leads to the expansion and bending of the actuators, aiming to provide hand function assistance while performing ADLs.

#### On-Board Power Supply Unit

A custom battery pack is in active development to power up the NeuroSuitUp device. The pack will be divided into modules. Each module consists of five Li-Po cells in series connection. The total open circuit voltage is approximately 22 V. Most of the device sensors however have a voltage requirement of either 5 V or less. Hence, a few step-down back converters are required to power these sensors. The output voltage of the step-down converters is programmable so the battery pack can supply any additional device. Furthermore, the capacity of the battery pack is adjustable. Connecting two or more battery modules in parallel connection can increase the total capacity of the pack. The initial design requirements demand that the NeuroSuitUp device can be independently in-use for an hour. By calculating the power requirements of the system, a battery pack with over 2000 mA capacity is considered adequate.

In a multi-cell battery pack, events such as over/under voltage, over-current and high temperature increase of a power supply can heavily damage the pack itself, the wearable device and, most importantly, injure the user. As a result, a battery management system (BMS) is required to prevent these conditions. The BMS of the battery pack is controlled by a micro-controller unit (MCU). The MCU can collect data with the use of diagnostic hardware from every battery cell and proceed to a predetermined action if necessary. Finally, the MCU can calculate the State of Charge, State of Health, and control the cell charge balancing [31], which can further protect the NeuroSuitUp device.

### 2.2. Sensors

#### 2.2.1. MARG

In order to acquire an accurate representation of the upper-body kinematics, keeping in mind the complex bio-mechanics of the human joints, three MARG sensors (Adafruit Precision NXP 9-DOF Breakout Board − FXOS8700 + FXAS21002) are used on each side (Figure 2), located at the segments of the *clavicle*, the *brachii*, and *antebrachii*. This allows for monitoring of the absolute orientation of each segment, and for the extrapolation of the state variables of the system (joint angles).

These sensors generate a digital I^2^C signal, composed of 6-dimensional inertial (accelerometer and gyroscope) and 3-dimensional magnetic measurements. Prior to usage, the individual sub-sensors require a calibration, to account for sensor drift, for the inertial sensors, and Hard and Soft Iron distortions for the magnetometer. The calibration sequence was conducted using the MotionCal tool, provided by the manufacturer.

Due to restrictions in available I^2^C addresses by the manufacturer, a multiplexer is introduced to allow for multiples of the same device to operate on the system. The generated 9-DoF (Degrees-of-Freedom) measurements are then forwarded via USB to a parser script that connects it to the ROS middleware package, which in turn handles filtering and distribution.

#### 2.2.2. Surface Electromyography

In order to connect the kinematic measurements of the system, to the physiological state of the wearer, commercial sEMG sensors were employed (MyoWare Muscle Sensor), at the main actuating muscle groups of the kinematic chain. For the shoulder joint, three sEMG sensors are connected to the *Pectoral Majoris*, *Deltoid* and the *Trapezoid*, and for the elbow joint, two EMG sensors are placed in the *Bicep* and *Tricep* muscle groups (Figure 2). Those groups were selected to effectively monitor the *agonist–antagonist* interactions that facilitate the control of the joint angles.

The sEMG sensors generate a continuous analogue signal, in the form of either a raw signal, or as in this application, an amplified, rectified and linearized EMG signal, referred to as an EMG envelope. This type of linear data format was selected to remove negative voltage values from the measurements and reduce noise and processing overhead further down the data processing line [32]. In order to achieve the high density of measurements required for the effective monitoring of muscle activity, the sensors are split into the analogue inputs of two Arduino Mega 2560, to avoid bottlenecks in the ADCs (Analogue-to-Digital Converters) of the breakout boards.

#### 2.2.3. Flex Sensors

Flex sensors are used to determine the level of bend in an object. They can be utilized in biomedical devices to record both static and dynamic positions on the human segment [33]. Despite desirable characteristics such as robustness, affordability, and lifespan, they usually display a nonlinear response and lower sensitivity at small bending angles. Regarding the different types of sensors of the MSS, four Flex sensors with two SparkFun Qwiic Flex Glove Controllers were used for measuring the movement of the fingers. Each one of them was put on the index, long, ring, and small fingers (Figure 2).

All communication with this controller is done purely through I^2^C, using the easy Qwiic technology, making it even easier to use. The Qwiic Flex Glove Controller includes an on-board ADS1015 ADC to I^2^C chip, allowing it to receive a variety of analogue inputs without touching the microcontroller’s ADC pins.

#### 2.2.4. Pressure Sensors

Flexible pressure sensors can provide sensory capabilities to robots, prosthetics, and other technologies. Two RFP602 Resistive Thin Film Pressure Sensor—Force Sensing Resistor 5KG membrane pressure sensors are attached to the thumb and middle finger tips (Figure 2), capable of sensing static and dynamic pressure on any contact surface, and the pressure applied to the sensing region of the sensor is converted into a resistance signal by the membrane pressure sensor. The external pressure variations are then recorded using the force-resistance calibration curve. The lower the resistance of the sensor output, the higher the pressure.

In order to test the responsiveness of the pneumatic actuator, another pressure sensor was added between the top of the middle finger and the associated pneumatic actuator for the purposes of the experimental validation. This pressure sensor is a DFRobot Pressure Sensor—SEN0298 with a pressure measuring range between 30 g and 1.5 kg.

### 2.3. Actuators

#### 2.3.1. EMS

The NeuroSuitUp platform was designed, from its conception, with the capability to support users with varying levels of mobility. This necessitates the implementation of an external actuation layer, which would enable the movement needed for a successful rehabilitation regimen. In the literature, that solution would usually entail some form of an exoskeleton [15,24], which would use a form of external motors to actuate with highly repeatable and powerful movements. This paradigm, however, carries the risk of passivity on the side of user, with the possibility of limited muscle activity, which can be considered critical in any rejuvenation attempt, post-SCI or stroke [34]. This has been noted and accounted for in many works in the past few years, most notably through the introduction of electrical stimulation of the muscle tissue for locomotion, which comes with its own limitations, specifically with regards to extensive use, due to muscle fatigue onset [35,36]. In our implementation, a hybrid system is being developed, in order to ensure both the high repeatability of external motor actuation, and the physical engagement of electrical muscle stimulation. Such systems have been devised in the past, mainly through the use of clinical-grade FES stimulation [37,38]. The electric stimulation of our system utilizes a commercial EMS device (Sanitas SEM 43), activated through a Proportional-Derivative (PD) controller that calculates and attempts to minimize the error between the expected (by the exercise-at-hand) and the real-world body pose. The device can output a max current of 200 mA, at a frequency up to 150HZ, and generates a biphasic rectangular pulse (https://sanitas-online.de/en/p/sem-43-digital-ems-tens-1/, accessed on 1 March 2023).

#### 2.3.2. Pneumatics

Pneumatic actuators are commonly used in neurorehabilitation to improve motor function and range of motion in patients functional deterioration of the hand [39]. Pneumatic Artificial Muscles (PAMs) [40,41] are soft pneumatic actuators that typically consist of a deformable inflatable chamber reinforced with polymeric fibres, inextensible fabrics, or stiff rings [42]. These actuators are lightweight and flexible, making them suitable for use in wearable devices such as exoskeletons and soft robotic gloves [43].

The Whitesides Research Group at Harvard (https://gmwgroup.harvard.edu/, accessed on 1 March 2023). is credited with the initial development of the pneumatic networks (PneuNets) class of soft actuators. A soft PneuNet is described by the movement, expansion, and bending of the soft structure of the actuator, assisted by a fluid (e.g., compressed air) [44]. Most pneumatic actuators are made of two layers, where the external layer is the expandable one and the inner layer is the inextensible one [45]. Through the air chambers of the external layer, the compressed air flows and so the soft walls expand and inflate [46].

Many researchers have been interested in the bending behaviour of PneuNets actuators in recent years, and as a result, numerical, analytical, and empirical (statistical) models have been constructed to emphasize the relationship between bending angle and input pressure [47]. Soft robotic gloves with PneuNet actuators may enable flexible and adaptive movements, allowing patients undergoing rehabilitation to make bending motions that safely adhere to human finger motion [48]. In the present study, the utilization of silicone-based soft PneuNet actuators in a soft robotic glove is being explored as a viable solution for the neurorehabilitation of the human hand.

### 2.4. Middleware Layer

The connective tissue between the Hardware and the Serious Gaming Application lies in the Middleware layer of the system architecture (Figure 1). Physically, it is represented by the main Linux computer in which the Arduino devices are connected. The operation occurs exclusively within the ROS ecosystem, both for Input/Output (I/O) handling, Data Management (filtering, utilization, and transfer) and data acquired by the SRG MSS for the control and distribution of compressed air to the PAS.

The data flow within the layer can be divided into three stages. Initially, sensor data is parsed from the USB port, via a Python script operating as a node in the ROS ecosystem. The said data then becomes available for manipulation. In the case of the MARG sensor data, specifically, an implementation of Madgwick’s filter for 9-DoF is used [49], for each 9 × 9 data-frame (3 × accelerometer, 3 × Gyroscope, 3 × Magnetometer) generating a unit quaternion for each of the 6 MARG sensors, representing the changes in orientation of the kinematic chain segments.

Alongside the parsed sEMG, the data is sent to a RosBridge-based server, running native within the ROS package.The information then becomes available to the Serious game application, with the Quaternions being applied to the corresponding kinematic segment of the player model, generating live movement, and the sEMG being used in the calculation of the exercise success visual representation sphere, which can be seen Figure 3a.

Lastly, the ROS package handles the calculations for the actuator controllers. By comparing the wearers live kinematic state, with the state expected by the rehabilitation regiment, an error value is generated, which is then minimized via activation of the EMS devices, located in the main actuating muscle groups of the aforementioned kinematic chain.

### 2.5. Application Layer

It is a truism that gaming is fun. The engagement potential of gaming led to its incorporation to serious uses when user engagement is desired or required. In that context, serious applications for gaming have been developed for several purposes. These include secondary education, especially for STEM topics [50,51], healthcare education [52], and wellness and healthcare interventions [53,54,55,56]. All these “soft” implementations of serious games come in contrast to one of the more “hard” applications of it in the form of neuro-rehabilitation interventions [57]. A recent review [58] has identified more than 27 publications regarding game-based stroke neuro-rehabilitation in Extended Reality (XR) alone. It is a field where contemporary modalities are finding fertile ground due to their increased immersiveness potential.

The HEROES-NeuroSuitUp approach implements a serious game that combines engagement features prolific in all serious games (meaningful play, handling failure, varying difficulty, and meaningful feedback–corrective feedback (cf) review). In addition it incorporates biofeedback principles both audio-visually and through closed feedback loops with biosensor-computer interfaces such as electromyogram and neuro-stimulation wearable devices. The HEROES serious game intervention is based on biofeedback principles, deployed in a playful context. The game puts the rehabilitating user in the shoes of an aspiring martial artist. In a virtual dojo, the user must precisely follow the motions of a digital “sensei”, the guide that will be demonstrating the motions of the users extremities (Figure 3a).

Technically, the game comprises a front end, developed in Unity, where all interaction with doctors, physical therapists, and patients interact and a back end, a MongoDB deployment, which includes all exercise information. This information includes users and their roles (doctor, physical therapist, patient), exercise information and patient progress. Exercises are organized in a hierarchical structure, with “*Pose*” being the most basic component and involving the angles that describe the configuration of they user’s limbs in a particular exercise stance. Two consecutive poses, between which the user must move to complete it, is a “*Task*”. Consecutive tasks comprise an “*Exercise*”, which is a complete training session for the patient. Finally, consecutive exercises constitute a “*Level*”, which includes all therapeutic regimens of exercises that the patient will need to complete for support of their rehabilitation program.

User input is provided by the HEROES wearable jacket. Through the ROS bridge and the ROS# unity package, direct access to the angles of the patient’s extremities is available to the unity 3d model. The data, in the form of unit quaternions, are first converted into the form of Euler angles, representing the World Frame transformations with regards to the Game world. These transformations are then applied to the associated segment of the kinematic chain, and the corresponding joint angles are calculated from the Rigid Body transformation between adjacent segments. Finally, in order for the Game World coordinate transformations to match the real world movements performed by the user, a calibration procedure is performed at the start of each training session, where the user is guided through performing transitions between specific poses (arms relaxed by the sides of the body, arms extended to the sides, and arms extended forward). This procedure allows for robustness of calibration through slight adjustments in the initial orientation of the segments prior to operation. Through this procedure the calibration is user specific but not task-specific and can carry over different motor tasks performed by the same user.

The front end of the HEROES application includes the level, exercise selection, and training windows. This application is deployable in desktop computers and portable to VR headsets such as MS HoloLens and Oculus quest.

Creating exercises is done in the level window. In the level window (Figure 3b) the doctor, or physical therapist, has access to a human-like mannequin that can be manipulated in its joints, so that it can take any physiologically relevant pose. The user can input joint angles both interactively, by visual manipulation of the mannequin and by direct numerical input. The user can select from several speed settings for each task so that the same motion can change in difficulty. In the exercise selection window, one of the windows two accessible to the patients, the user is presented through their personalized level in the form of different martial arts belts. Each “belt” includes exercises so that the whole exercise can be narratively translated as a martial artist’s journey, for motivational purposes. (Figure 3c).

When selecting a belt, the user is presented with the training window (Figure 3a) in which two human-like mannequins exist. One, the “sensei”, is programmed to demonstrate the exercise and, after a readiness cue, start executing it while waiting for user input. User input, provided by the input from the HEROES wearable is displayed in the second mannequin that represents the user. In order for the user to achieve a perfect performance they have to move their hands exactly in the position and at the same time that the sensei is also moving them. Both a speed and a position match check is made for each joint to ensure that the task is executed correctly. To adhere to good biofeedback practices, a series of audiovisual cues are provided to facilitate feedback for the patient. At the bottom of the screen a sphere is constantly reflecting, its size indicating the performance of the user. In addition, as the task evolves, the user’s mannequin is filling from top to bottom with a silver shade to demonstrate its progress. Additionally, at the same time, the user mannequin is filling up with a golden shade according to the user’s progress with the task. That way the user can implicitly understand when to put in extra effort, similar to all biofeedback implementations, and facilitate their own rehabilitation efforts. Standard gamification techniques are also applied with the score increasing at the rate of exercise adherence, for each time. At each score milestone, word cues («great», «excellent», etc.) are provided to further motivate the patient. Concluding, the editing capacities, gameful experience for the patient and biofeedback principles implemented, make the HEROES serious game the hub of user interaction in the complete HEROES rehabilitation solution.

## 3. System Validation

### 3.1. On-Board MARG Sensor Validation through the Use of Stereoscopic Vision

For the validation of the wearable system’s movement tracking, we made use of a secondary system to obtain paired measurements, alongside the main system. For this purpose, an Infrared and Depth Camera system was selected. These systems have been used for facial and skeletal tracking in many applications, such as gaming, security systems, as well as medical rehabilitation [59]. In medical research, applications using Microsoft Kinect adjacent systems are used to track upper and lower-body joint movements for rehabilitation treatments, mainly for the validation of the correct execution of rehabilitation regiments by patients with motor disabilities [60]. Another use-case of a Kinect-like system lies with the measurement of gait parameters, either for elderly care, or for patients with movement difficulties. The purpose of the use of the selected system in our use case can be considered to be similar. A doctor or a physiotherapist tracks the muscle movements as a whole and measures the angles and the extent of the body extremities [61].

An Orbbec Astra Pro infrared and 3D depth camera system is utilized in the current implementation to track the movement of the participants’ joints in 3D space. During the execution of a set of representative exercises, the motion data generated by the wearable device is compared with measurements from the visual system to validate the accuracy of the wearable system.

The first step for detecting and tracking the users’ movements from depth-sensor data is the definition of a human model, which can interact with the coordinate system predefined by the Unity Game Engine [62]. The manufacturer provided Nuitrack SDK, which comes in the form of a Unity package, and contains pre-made C# Objects that have been matched to the main segments and joints of the kinematic chain of the human body.

In order to track the angle of a joint using a camera setup, the rotation of the 3D dummy part containing said joint must be calculated. In practice, this is achieved through the use of a class method in C#. The method uses the data inputs from these body parts and calculates unit Quaternions from the Euler angles provided by the NuiTrack SDK, for each 3D dummy part involved in a movement, based on the following transformations:$$q_{x}=sin\left(R/2\right)\times cos\left(P/2\right)\times cos\left(Y/2\right)-cos\left(R/2\right)\times sin\left(P/2\right)\times sin\left(Y/2\right)$$
$$q_{y}=cos\left(R/2\right)\times sin\left(P/2\right)\times cos\left(Y/2\right)+sin\left(R/2\right)\times cos\left(P/2\right)\times sin\left(Y/2\right)$$
$$q_{z}=cos\left(R/2\right)\times cos\left(P/2\right)\times sin\left(Y/2\right)-sin\left(R/2\right)\times sin\left(P/2\right)\times cos\left(Y/2\right)$$
$$q_{w}=cos\left(R/2\right)\times cos\left(P/2\right)\times cos\left(Y/2\right)+sin\left(R/2\right)\times sin\left(P/2\right)\times sin\left(Y/2\right)$$
where R: Roll—rotation in x-axis, P: Pitch—rotation in Y-axis, Y: Yaw—rotation in z-axis.

From these transformations, the Quaternion Rotation Matrix can be generated as follows:$$RotationMatrix=\left[\begin{array}{ccc} 1-2(q_{y}^{2}+q_{z}^{2}) & 2(q_{x}q_{y}-q_{w}q_{z}) & 2(q_{w}q_{y}+q_{x}q_{z})\\ 2(q_{x}q_{y}+q_{w}q_{z}) & 1-2(q_{x}^{2}+q_{z}^{2}) & 2(q_{y}q_{z}+q_{w}q_{x})\\ 2(q_{x}q_{y}+q_{w}q_{y}) & 2(q_{w}q_{x}+q_{y}q_{z}) & 1-2(q_{x}^{2}+q_{y}^{2}) \end{array}\right]$$

The program then calculates the difference between the Quaternions in the desired axis between two adjacent 3D dummy body parts and converts the result back to Euler angles, based on the following transformations:$$\left[\begin{array}{c} R\\ P\\ Y \end{array}\right]=\left[\begin{array}{c} atan2(2\left(q_{w}q_{z}+q_{x}q_{y}\right),1-2\left(q_{y}^{2}+q_{z}^{2}\right))\\ -\pi /2+2atan2(\sqrt{1+2(q_{w}q_{y}-q_{x}qz)},\sqrt{1-2(q_{w}q_{y}-q_{x}qz)})\\ atan2(2\left(q_{w}q_{x}+q_{y}qz\right),1-2\left(q_{x}^{2}+q_{y}^{2}\right)) \end{array}\right]$$

The output of both the visible-light camera and the infrared camera is displayed. This allows the researchers to observe the tracked joints at all times. The precise location of the joints that Nuitrack SDK defines as the centre of each joint can be tracked and whether the lighting or position of the person tracked by the device is adequate can be determined. The accuracy of this type of depth cameras ranges between 2 and 4 mm when the subject stands between 0.5 and 3.5 m from the camera and the room lighting is adequate [63].

For the validation of the MARG system, a 3 exercise experiment was devised, focusing on movements around the main rotation angles of the upper body kinematic chain. Each exercise was repeated 10 times by all 10 of the participants and included concurrent measurements from the NeuroSuitUp platform and the aforementioned NuiTrack Camera setup, each running on a separate application of the Unity Game Engine (Table 1).

### 3.2. Soft-Robotic Glove Experimental Setup

The human hand is composed of a variety of different bones, muscles, and ligaments that work together to allow for a wide range of movement and dexterity. Motor control of the human hand is required to preserve independence throughout daily tasks. However, neurological disorders may have a direct impact on the usual prehension patterns (grasping, wrapping, and pinching), resulting in hand functional impairment [64].

To evaluate the functionality of the SRG, several validation experiments were conducted in order to ascertain the development of the device. Specifically, the experimental subjects demonstrated no muscular or neurological hand impairments.

Each experiment consisted of two sessions. During the first one, the SRG actuation system is not activated (Actuator State = 0), since the healthy performance is tested. During the second session, the actuation system is activated (Actuator State = 1) in order for the subject to perform the ADLs with the SRG assistance. The subjects were recruited to perform three (3) simple ADLs during each session (Table 2) while wearing the SRG. Sensor data, actuation status (0 indicates that the PAS is off, 1 indicates that the PAS is on), and timestamps for each second during the sessions were recorded in a CSV file for each participant.

### 3.3. Participants and Questionnaires

In total, 10 healthy participants, 5 male and 5 female (median age = 26 years, interquartile range (IQR) = 4, first quartile (Q1) = 24.5, third quartile (Q3) = 28.5), participated in the system validation trials of the platform that were conducted during December 2022 and January 2023. Informed consent was obtained from all participants involved in the study, in accordance with the institutionally approved and published study protocol [27]. The participants had no prior experience with wearable robotics and rehabilitation experiments or the use of serious games. The participants completed the tasks previously described that took place in the Thess-AHALL Living Lab [65] of the Medical Physics and Digital Innovation Laboratory site and then completed a series of questionnaires to evaluate the perception of robotics, mental effort, and discomfort, as part of the User Experience. Those included the GODSPEED Robotics Questionnaire [66,67], the Subjective Mental Effort Questionnaire (SMEQ) [68], and the Locally Experienced Discomfort Questionnaire (LED) [69].

### 3.4. Statistical Analysis

Statistical analysis of the data gathered during the system validation trials was performed according to the planned analysis, which has already been described in the published study protocol [27]. Analysis was performed in Python and the statistical significance level was set at 0.05. In short, continuous variables were explored for normality by means of the Shapiro–Wilk test and the appropriate descriptive statistics (parametric/non-parametric) were used accordingly for reporting descriptive statistics. Continuous variables with a normal distribution were reported as the mean (SD), while variables without a normal distribution were reported as the median and interquartile range (Q1–Q3). Possible associations between variables were investigated using the Spearman correlation coefficient.

## 4. Results

### 4.1. Robotic Jacket Experimental Results

The experiments generated two sets of data for each exercise, one from the stereoscopic camera and another from the serious game application. The main issue with the data analysis process was the fact that the sampling rate for each system was different (camera at 1 Hz, SG at 5 Hz). Furthermore, the fact that both systems ran from two different applications compounded the issue of temporally matching the data.

To account for the synchronization issue, timestamps, which, since the two data collection applications ran on the same computer were the same, were used for indexing the two datasets. In order to accommodate the differing sampling rates, downsampling was implemented on the dataset with the higher rate, using mean values. Finally, to compare the two paired waveforms, Spearman’s Correlation coefficient (Table 3) was calculated within the timeframe of each exercise (Figure 4 and Figure 5).

### 4.2. Soft Robotic Glove Results

The strategy for data exploration and analysis of the sensor data collected when performing the three tasks mentioned at the SRG experimental setup is based on hypothesis testing. We hypothesize that the sensor data differ when the actuator is “off” (indicated by 0 in the data) versus when it is “on” (indicated by 1 in the data). To test this hypothesis, we compare the distributions of the sensor data between these two states.

One method is to compute and compare the standard deviation of the sensor data for each state. The standard deviation quantifies how far the data deviates from the mean or expected value. The data recorded for each participant was loaded and concatenated into a single dataframe using python. The data was then divided into two categories based on the actuator status. The standard deviation of each sensor measurement was then determined, and the findings are visualized in the bar charts of Figure 6. The x-axis of each plot depicts actuator status, while the y-axis represents sensor data standard deviation.

The glove MSS SD data compared on both actuation states, in most cases, display non-statistically significant differences during the experiment tasks, which provides useful insight into the efficacy of the SRG neurorehabilitation aid (Table 4). Furthermore, the SD data of the pressure sensor positioned between the top middle finger and the associated pneumatic actuator display a significant difference between the actuation states in most cases. This demonstrates the relevance of the pressure sensor data in providing a more comprehensive assessment of the actuation and emphasizes the need of using multiple sensor data in evaluating the success of the assistance.

### 4.3. User Experience

The participants in the system validation trials perceived the usage of the combination of robotics and serious game platform mostly positively, as they reported a mean total Godspeed score of 90.9 out of a possible 120 maximum score (SD 12.09). All Godspeed subcategories were also reported positively. In detail, the mean Anthropomorphism was 16.6 (SD 3.47; maximum 25), the mean Animosity was 20.4 (SD 5.56; maximum 30), the mean Likeability was 22.7 (SD 2.16; maximum 25), the mean Perceived Intelligence was 19.5 (SD 3.50; maximum 25), and the mean Perceived Safety was 11.7 (SD 3.91; maximum 15). Figure 7, depicts all the reported total Godspeed scores by the participants, as well as the mean scores by subcategory.

The participants did not experience much difficulty during the trials of the reported tasks, as the median SMEQ rating was 10 (Q1 = 5, Q3 = 10), corresponding to the answer “not very hard to do”. Only one participant rated the effort with a 20 in the SMEQ scale. Similarly, almost no local discomfort was experienced by the participants during donning the wearable robotics jacket and gloves and participating in the validation trials. In detail, only the right shoulder-arm was reported with a median LED rating of 1 (Q1 = 0, Q3 = 1) out of possible 10 maximum, corresponding to “hardly any complaints”. All other body areas were reported with a median LED rating of 0, corresponding to “no complaints at all”. The highest reported LED rating by any participant was 3 (between “some complaints” and “quite a lot of complaints”) for the shoulder-arm area. Figure 8, depicts all reported SMEQ answers and all LED answers of the participants for the different body parts. No significant correlations were revealed between questionnaires.

## 5. Discussion

### 5.1. Key Findings, Limitations, and Future Development

Regarding the validation of the robotic jacket through the use of a stereoscopic camera, we have shown that the selected approach of utilizing segment-orientation using MARG has performed adequately for the majority of the use-cases. Correlation between the joint angles measured by the camera and the angles computed by the MARG in the case of the robotic jacket was above 0.85 in 21 out of 30 arm exercises in total performed by the participants. In more detail, the only exercises that did not reach above 0.85 correlation were the shoulder sagittal abduction performed by all 10 participants. In this exercise, the participant moves their arm in the sagittal to the camera plane (moving forward and closer to the camera at first and then returning to the neutral position). In this exercise, only three participants were measured with a correlation of camera-to-MARG angles above 0.70 (participant 02: 0.73, participant 01: 0.84, participant 03: 0.86) while in the rest, correlation was not achieved. In contrast, in both the two other exercises (shoulder coronal abduction, elbow flexion/extension), where the movement of the arm was perpendicular to the camera, the correlation ranged from 0.85 to 0.99, suggesting that this is due to limitations of the camera rather than the accuracy of the MARG system. To resolve this issue we aim to use a multi-camera setup from different points-of-view (frontal view and side view) to implement a vector computation of correlation to validate the MARG sensor subsystem’s performance in different angles of arm movement during exercises. Furthermore, we could argue for the implementation of two additional MARG sensors, placed at the base of the neck, to enable a more accurate and standardized in-game calibration process for each user in real-time, and the lower back, near centre mass of the body to provide an accurate image of the body pose.

Regarding the validation of the soft robotic glove through comparing sensor data between active user grip and passive user grip (deactivated Actuator State (0) and activated Actuator State (1), respectively), while no significant differences were observed in the two states, a further breakdown is necessary. The SDs of the four flex sensors and the three pressure sensors were comparable in all three grip exercises, albeit somewhat lower in the passive user grip. This demonstrates the ability of the soft robotic glove to generate enough force to perform basic ADLs. Furthermore, in the cylindrical and the lumbrical grip exercises, the SD of the sEMG values were non-significantly lower in the Actuator State (1), denoting the actual less burden of neuromuscular activation required by the participant in this state. On the other hand, our experience with these validation experiments showed us that real-time fine control of pneumatic actuator fluid pressure is necessary to both avoid damage to the actuators during more strenuous ADLs and to allow for fine control, fine movements, and a more meaningful exercise regimen design for patient participants.

Overall, and in regards to both robotic devices (jacket and glove), our validation experiments provided valuable insights and contributed to the overall research effort towards the next development cycle for an upper body, arm, and hand neurorehabilitation/assistance robotic garment. Among these insights we can emphasize the need for a more robust hardware design, including device stability on-board the user. While no difficulty to use and no discomfort in general was reported by the participants in the validation trials (and in that sense even less difficulty to use and complaints when compared to the pilot testing reported in Ref. [27], these participants were healthy individuals performing standardized simple repetitive tasks. In order for the platform to be considered similarly unobtrusive by the patient populations that it is primarily aimed at (chronic SCI and chronic stroke), design improvements will be implemented in the next development cycle and prior to the commencement of the clinical trials. Among these improvements, a more cost-effective approach to sEMG electrodes needs to be considered. In future measurements, we will also be able to detect the air pressure flowing into the actuators while employing the soft robotic glove in ADLs by integrating a pressure sensor with the air pump that inflates the actuators. We will also use a dynamometer to measure the grasping force in all directions during these studies. Further validation experiments, with more exercises, increased exercise complexity, and a bigger sample of healthy participants is considered necessary before reaching the clinical trial status. Overall immersion is aimed to be improved through the implementation of the serious game in an Augmented Reality (AR) module [70]. Finally, signal processing pipelines with the intention of both offline analysis for gathering more insight, and online/on-board techniques to improve device accuracy and efficacy are among the key next development steps that we aim to undertake [71].

### 5.2. Related Work

Recently, wearable robotic garments—including gloves—have been used for neural rehabilitation in both research and clinical settings, aiming to assist in various scenarios such as recovery from stroke, spinal cord injury, and mitigating the effects of cerebral palsy [72]. Such devices need to be personalized to each patient’s body shape and needs, as well as account for variations in daily installation on the body; hence, they need fast-processing control systems, which can be adapted quickly, despite their complexity [73]. Consequently, solutions based on artificial intelligence (AI) algorithms have emerged, leveraging similar research and development, aimed at improving control algorithms for mobile robotic exoskeletons. The majority of projects reported in the literature to date involve active exoskeletons and similar wearable robotics garments made out of a mixture of hard materials (mostly plastic and metal), capable of a limited number of degrees of freedom (generally fewer than 4), while being controlled by algorithms that use Artificial Neural Networks (ANNs) for quick adaptability [74].

The increasing interest in the development and adoption of serious gaming for neural rehabilitation purposes constitutes a valuable technological convergence with respect to wearable robotics, as a potential new component in both research and clinical practice. Specifically, the use of serious gaming in neural rehabilitation for motor dysfunctions has been intensively researched during the past three decades [75].

VR and AR technologies can be utilized as a visual aid for performing a task or immersing oneself in a different environment [76]. A BCI paradigm in VR has also been suggested, which allows motor priming for patients with limited motor control [77]. This leads to the exploration of the employment of VR/AR systems in methods for rehabilitation therapies, although these technologies are still in their infancy and require further research [78]. The combined use of these technologies can potentially motivate patients to optimize their training and therapeutic neural rehabilitation routines, improving the outcome.

### 5.3. Clinical Scope

Neurorehabilitation interventions have exploded in the last 20 years. Nowadays, it is believed that the brain has regenerative and dynamic reorganization potential, months and even years post-damage [79]. While interdisciplinary rehabilitation interventions are assumed to be the core of neurorehabilitation, one of the key specialities is physiotherapy, which is primarily aimed at restoring activities of daily living (ADLs), with goal to improve the function of walking and recovery of balance and movement [80]. Various rehabilitation techniques have been utilized by physiotherapists to assist the patient’s recovery, including conventional physiotherapy, electrostimulation, exoskeleton or robot-aided therapy, virtual reality (VR) therapy, serious games-based therapy, or a combination of these [81]. Conventional physiotherapy methods and approaches include the Bobath method, Brunnström method, Rood method, and proprioceptive neuromuscular facilitation approach [82]. In addition, in the last decades the motor learning approach was proposed. Motor learning is based on intensive, repetitive task-specific training, which involves the active practice of task-specific motor activities [83]. Concepts and approaches related to motor learning include repetitive training, mirror therapy, impairment-oriented training, task-oriented therapy, constraint-induced movement therapy, mental training and movement observation, functional electrical stimulation, neuromodulation, treadmill training, and rhythmic acoustic stimulation [82]. Although most of these neurorehabilitation approaches are effective, none appear to be superior to any other. Neurorehabilitation, using a mix of different approaches tailored to each patient, gives beneficial effects on functional recovery and independence [84]. Regarding robot-aided therapy, it is effective but has no significant advantage over conventional physical therapy [85]. A systematic comparison of different approaches suggested that robot-aided therapy is among the most effective techniques for neurorehabilitation [86], and a combination with conventional physiotherapy is more likely to achieve independent walking than physiotherapy alone [87].

Patients with neurological disorders would benefit from technological interventions that may increase their functional ability and independence. Neuromuscular electrical stimulation (NMES) can selectively generate movements from the patient’s own muscles to potentially restore their function [37]. NMES can be delivered as fixed sequences of stimulation for exercise and retraining purposes (therapeutic electrical stimulation) or as functional electrical stimulation (FES) that aims to assist functional and purposeful movements. This is achieved by applying electrical stimulation to muscles that are involved in specific functional activities such as standing, walking, reaching, and grasping. A FES system that facilitates a specific activity is often called a neuroprosthesis [88]. Numerous applications of NMES/FES have been developed successfully to assist movement of the upper and lower extremities [89]. Lower limb FES applications are not widely used, partly because of their cost and partly because it is unclear if they provide a superior improvement in activities of daily life when compared to conventional therapies and ankle-foot orthoses. However, there is evidence that FES may improve gait, balance, and range of motion [90]. Studies for upper limb FES applications have also reported encouraging results [91,92]. The use of NMES/FES in the rehabilitation of a patient with neurological disorders may not only have an adaptive effect (direct compensation for motor disability) but also a restorative effect. Evidence from recent systematic reviews and meta-analyses suggests that NMES/FES may promote restorative changes in several neurological disorders that are at least equivalent to changes promoted by conventional therapy [36,93]. The results suggest that many of the adaptations elicited by NMES/FES occur within the nervous system and the intervention may take advantage of neuroplasticity to restore the patient’s ability to perform voluntary movement [35]. In addition, there is preliminary evidence that NMES/FES may elevate serum levels of brain-derived neurotrophic factor (BDNF), a neurotrophin that plays a role in the expression of neuroplasticity [94].

## 6. Conclusions

Our work takes a systematic approach towards the development of a multi-modal platform of modular devices and applications, where all the underlying technologies and their optimal cooperation is considered, validated, and tested under the scope of a unifying system architecture. In the current manuscript, we presented the system architecture, development, and validation experiments of the NeuroSuitUp platform of wearable robotics jacket and gloves and serious gaming for the motor rehabilitation of patients with neurological disability, to be primarily utilized in two intervention-based clinical trials for SCI and stroke. Ten healthy subjects participated in system validation trials, performing three arm and three hand exercises and completing user experience questionnaires to measure difficulty to use, discomfort, and perception of robotics. An acceptable correlation was observed in 23/30 arm exercises performed with the jacket, demonstrating the validity of the MARG approach to approximate kinematic chain segment orientation, while providing useful sEMG information. No significant differences in glove sensor data during actuation state were observed. This demonstrated the ability of the soft robotic glove to generate enough force to perform basic ADLs, all the while lowering the burden of neuromuscular activation required by the participants. No difficulty to use, discomfort, or negative robotics perception were reported. The system validation trials provided adequate insights for the next development steps before implementing our approach in clinical trial setting. Design improvements during those steps will implement additional absolute orientation sensors through MARG sensors, MARG and sEMG sensor data fusion to be used as input to biofeedback for controlling the serious game, improving the overall immersion through the game implementation in AR, as well as fabrication improvements towards the general robustness of the wearable modalities.

## Figures and Tables

**Figure 1 sensors-23-03281-f001:**
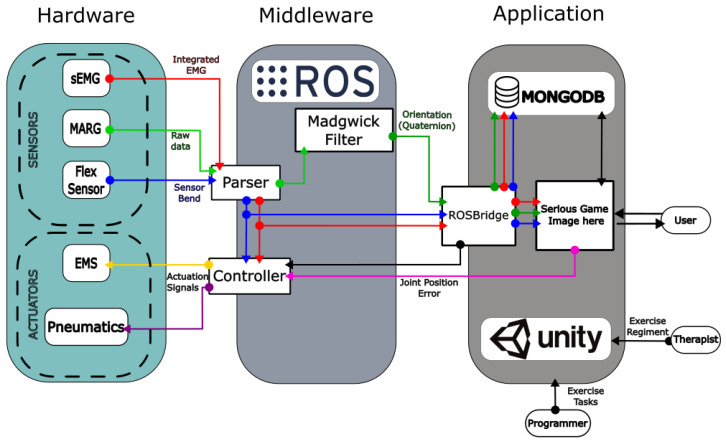
NeuroSuitUp system architecture.

**Figure 2 sensors-23-03281-f002:**
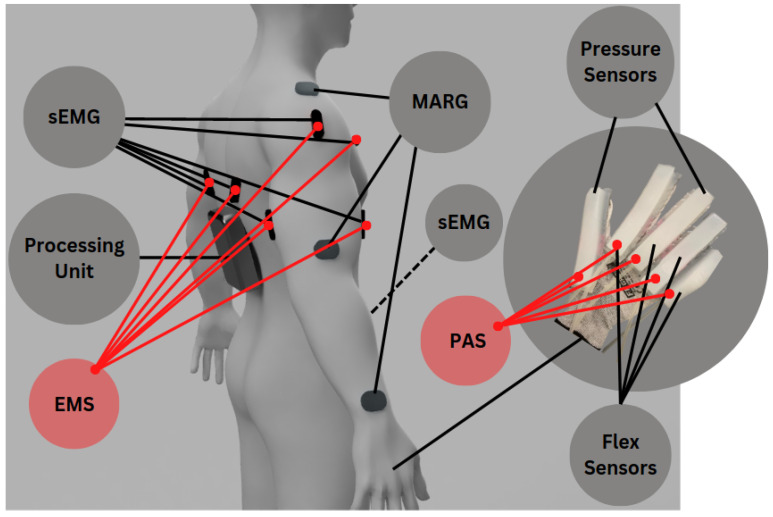
Concept image of the NeuroSuitUp platform sensor placement.

**Figure 3 sensors-23-03281-f003:**
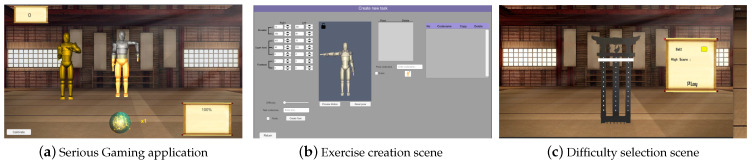
Serious gaming application for the NeuroSuitUp neurorehabilitation program.

**Figure 4 sensors-23-03281-f004:**
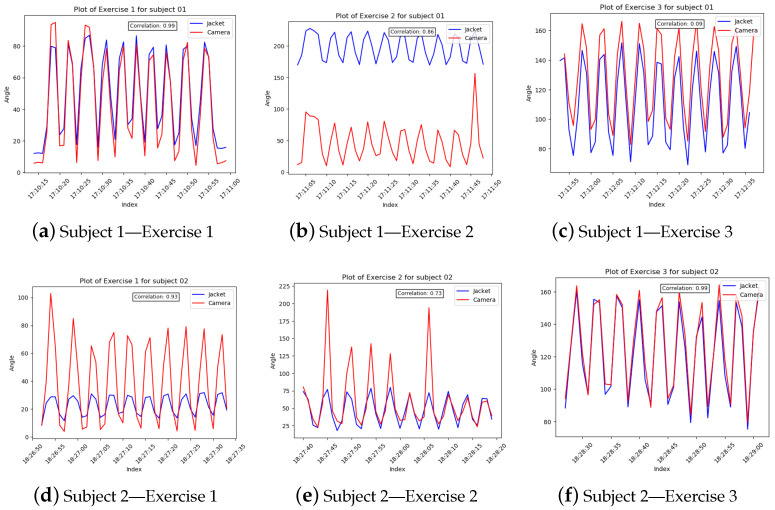
Robotic Jacket Validation Experiment Results for subjects 1 through 5. The camera angle measurements (red) are overlaid with the wearable jacket-generated inertial sensor approximation (blue) over time.

**Figure 5 sensors-23-03281-f005:**
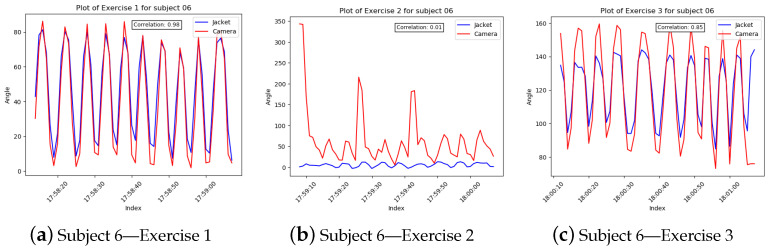
Robotic jacket validation experiment results for subjects 6 through 10. The camera angle measurements (red) are overlaid with the wearable jacket-generated inertial sensor approximation (blue) over time.

**Figure 6 sensors-23-03281-f006:**
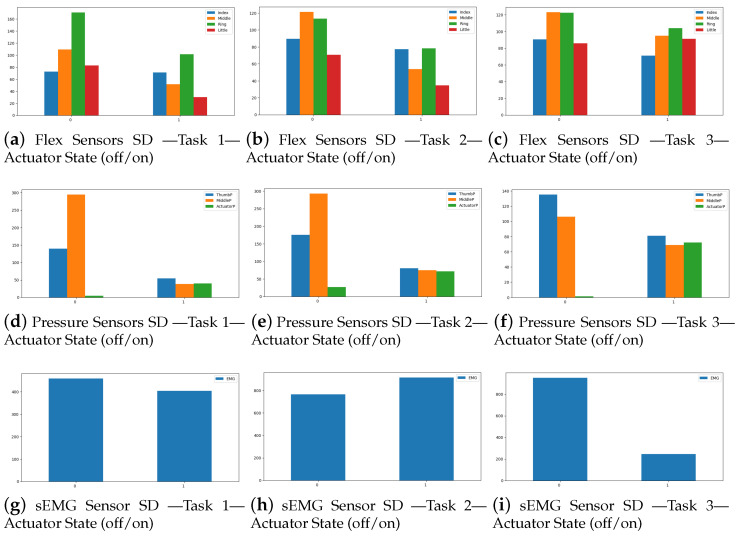
Soft robotic glove validation experiment results visualizing the standard deviation for each sensor between the Off and On actuator states during the three experiment tasks. More specifically, subfigures (**a**–**c**) indicate flex sensor SD results for the index, middle, ring, and little finger, respectively. Subfigures (**d**–**f**) indicate pressure sensor SD results for the thumb tip and middle finger tip and top, respectively. Subfigures (**g**–**i**) indicate sEMG sensor SD results of the extensor muscle activity.

**Figure 7 sensors-23-03281-f007:**
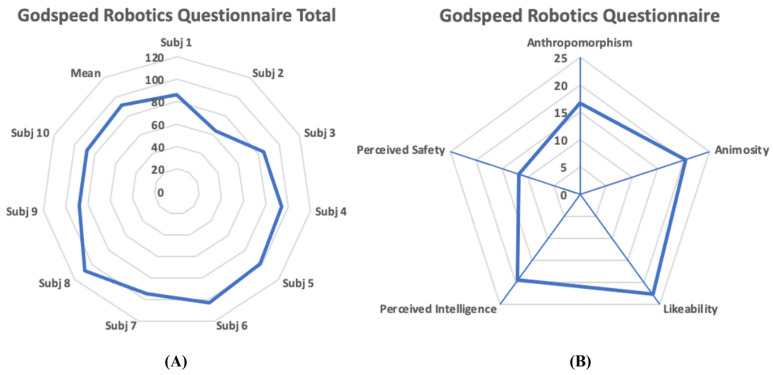
(**A**) Total Godspeed scores of all participants in the system validation trials and mean Total Godspeed score. (**B**) Mean Godspeed robotics questionnaire scores by questionnaire subcategory.

**Figure 8 sensors-23-03281-f008:**
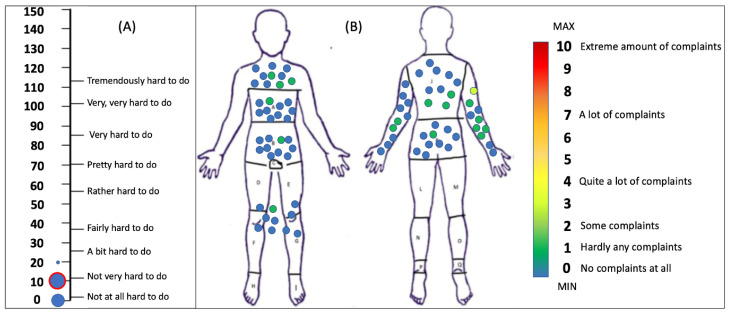
(**A**) All answers to the Subjective Mental Effort Questionnaire (SMEQ) by system validation trials participants. The size of the circles denotes the relative number of answers, while the red circle denotes median marking. (**B**) All answers to the Locally Experienced Discomfort Questionnaire (LED) by system validation trials participants were according to body area. Both legs are denoted as a single area, and so are the front and back surface of either arm. The colour inside the circles corresponds to the complaint intensity according to the colormap.

**Table 1 sensors-23-03281-t001:** Validation experiment—MARG sensor angles measurements.

Exercise	Description	Task	Figure
Shoulder Coronal Abduction	To measure the extension of the arm on the shoulder joint, along the Coronal plane of the wearer.	Starting from the position designated as $0^{\circ }$ extends and releases their arm in the coronal plane close to $90^{\circ }$ and holds for ≈2 s, before returning to $0^{\circ }$.	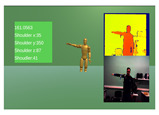
Shoulder Sagittal Abduction	To measure the extension of the arm on the shoulder joint, along the Sagittal plane of the wearer.	Starting from the position designated as $0^{\circ }$ extends and releases their arm in the sagittal plane, close to $90^{\circ }$ and holds for ≈2 s, before returning to $0^{\circ }$.	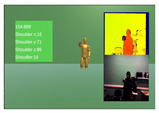
Elbow Flexion/Extension	To measure the flexion and extension of the lower arm on the elbow joint, along the Coronal plane of the wearer.	Starting from the position designated as $180^{\circ }$ extends and releases their arm in the coronal plane, close to $90^{\circ }$ and holds for ≈2 s, before returning to $180^{\circ }$.	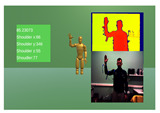

**Table 2 sensors-23-03281-t002:** Validation experiment—hand function activity.

Title	Description	Task	Figure
Cylindrical Grip	To assist the fingers curve around a cylinder shape, this grip combines extrinsic flexor action, lumbricals, and palmar interossei.	For ≈5 s the palm should contact the object with the thumb in direct opposition and abduction, then release. Wait for ≈10 s then repeat once again.	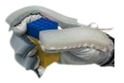
Spherical Grip	To curve around a circular item, the index, long, and ring fingers are abducted, while the thumb is opposed and abducted.	Grasp, squeeze for ≈5 s and then release a small ball. Wait for ≈10 s then repeat once again.	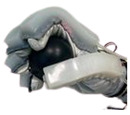
Lumbrical Grip	In this grip, the intrinsic muscles of the index, long, ring, and small fingers are most active, flexing the metacarpophalangeal joints to make contact with the object at the distal tips of the fingers and thumb.	For ≈5 s hold onto a flat object and then release. Wait for ≈10 s then repeat once again.	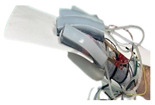

**Table 3 sensors-23-03281-t003:** Spearman’s Correlation coefficients for the wearable/visual validation tests.

Subject	Exercise 1	Exercise 2	Exercise 3
1	0.99	0.86	0.09
2	0.93	0.73	0.99
3	0.95	0.84	−0.68
4	0.91	−0.03	0.98
5	0.95	−0.19	0.96
6	0.98	0.01	0.85
7	0.99	0.59	0.90
8	0.98	0.53	0.90
9	0.91	−0.55	0.97
10	0.98	0.50	0.98

**Table 4 sensors-23-03281-t004:** Soft robotic glove validation experiment SD results.

Sensor	Placement	Task 1		Task 2		Task 3	
		Off	On	Off	On	Off	On
Flex	Index	72.37	70.82	89.53	77.39	90.72	71.13
	Middle	109.34	51.68	121.25	53.68	123.15	94.95
	Ring	170.56	101.27	113.26	78.32	122.62	104.04
	Little	82.67	30.21	70.57	34.79	85.84	91.33
Pressure	Thumb	139.96	54.47	175.09	80.59	135.21	80.87
	Middle Tip	294.67	38.37	292.70	75.04	106.14	68.74
	Middle Top	4.37	40.10	26.92	71.84	1.49	72.34
EMG	Extensor muscle	458.37	403.14	764.73	914.48	951.93	245.63

## Data Availability

Data from the NeuroSuitUp and HEROES studies will be made publicly available after the completion of the projects and will be accessible through their respective institutional project web pages under an Attribution-NonCommercial-NoDerivatives 4.0 International license. Data of the analysis presented in the current manuscript and the related code can be made available via a request to the authors following a Memorandum of Understanding (MoU) in the context of Open Research Initiative.

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
