# Peer review of "NeuroSuitUp: System Architecture and Validation of a Motor Rehabilitation Wearable Robotics and Serious Game Platform"

_sensors, 2023, doi:10.3390/s23063281_

Round 1

Reviewer 1 Report

This paper presents the system architecture, development and validation experiments of the NeuroSuitUp platform of wearable robotics jacket and gloves and serious gaming for motor rehabilitation of patients with neurological disability for the clinical trial of intervention spinal cord injury(SCI) and stroke. While primary idea of the paper is interesting, there are several issues and concerns that I found as a reviewer in the current manuscript.

Major comments:

1. It can be seen from Figure 1 and Figure 2 that the system mentioned in the article involves multiple sensor components, and there is also the problem of multimode fusion. The article does not provide enough information for the information collection and preprocessing of multimode fusion. Please provide further details about how to deal with the information fusion of multimode fusion.

2. In the sensor module in Chapter 2, A, is there any delay or lag in the data communication between sensors? B. Is the selection and calibration accuracy of MARG sufficient? Have alternative types (such as 9-axis IMU) been compared? C. Is the information from sEMG windowed? Looking forward to your more detailed description.

3. In the Application layer in Section 2.5, what is the relationship between the coordinate angle transformation of the game world and the angle transformation of the real world? Is calibration and error correction required? Can we adapt to different crowd tasks?

4. In the experimental verification of hand function in Table 2, we only evaluate whether it can be grasped. Will it involve the grasping force in all directions? Is there a corresponding evaluation index for effect verification?

5. In the discussion part, in the evaluation of 'the robot jack' and 'the soft robot globe', the real-time performance is mentioned, because the performance evaluation indicators of multi-sensor fusion are not really clear. Are there any comparisons with different teams?

6. Finally, the target of this article is the intervention spinal cord injury(SCI) and the stroke patients. Will more patients be considered for experimental verification in the future?

Minor comments:

1. Some of the diagrams in the text are not clear enough. Please provide clear and correct diagrams with reasonable labeling information. For example, there are problems such as incomplete text display in Figure 7.

2. There are several grammatical errors and typos in this paper. 

Ex.

a. Repetition of the phrase the rest of in “The on-board electronics are connected to the rest of the rest of the system through the use two Arduino Mega 2560s”;

b. Spelling errors, such as "expeeriments".

Author Response

Dear Editor and Reviewers,

The authors would like to express their sincere gratitude to the reviewers for their valuable comments and to the editor for this excellent opportunity to revise and significantly improve the submitted work. The changes in the revised manuscript attempt to clarify and meet all the suggestions behind the reviewers’ comments as well as to address all issues raised.

We sincerely hope that this new version of our manuscript will meet the reviewers’ expectations. In any case, we remain available to answer any new question. Please find our comment-by-comment replies in the attached file.

Reviewer 2 Report

See attached file for comments

Author Response

(The authors gave the same response as above.)

Reviewer 3 Report

The objective of this manuscript is to the system architecture and validation of the NeuroSuitUp body-machine interface (BMI). The platform consists of wearable robotics jacket and gloves in combination with a serious game application for self-paced neurorehabilitation in spinal cord injury and chronic stroke.

1- The motivations and contributions should be made clearly
2- The presentation should be improved. There exist some mistakes and typos.
3-Maybe it will be better to improve/add more tables with performance indices.

Author Response

(The authors gave the same response as above.)

Round 2

Reviewer 2 Report

Accepted and Congratulation to all authors.